# Seasonal influenza a virus lineages exhibit divergent abilities to antagonize interferon induction and signaling

Joel Rivera-Cardona[1], Neeha Kakuturu[1], Elizabeth F. Rowland[1], Qi Wen Teo[2,3], Elizabeth A. Thayer[1], Timothy J. C. Tan[4], Jiayi Sun[1], Collin Kieffer[1,2], Nicholas C. Wu[2,3,4,5], Christopher B. Brooke [1,2] *

1 Department of Microbiology, University of Illinois at Urbana-Champaign, Urbana, Illinois, United States of America, 2 Carl R. Woese Institute for Genomic Biology, University of Illinois at Urbana-Champaign, Urbana, Illinois, United States of America, 3 Department of Biochemistry, University of Illinois at Urbana-Champaign, Urbana, Urbana, Illinois, United States of America, 4 Center for Biophysics and Quantitative Biology, University of Illinois at Urbana-Champaign, Urbana, Illinois, United States of America, 5 Carle Illinois College of Medicine, University of Illinois at Urbana-Champaign, Urbana, Illinois, United States of America

* cbrooke@illinois.edu

**Data Availability Statement:** Single cell RNAseq dataset used here can be found at GEO accession number GSE272070. In house analysis used for

## Abstract

The circulation of seasonal influenza A viruses (IAVs) in humans relies on effective evasion and subversion of the host immune response. While the evolution of seasonal H1N1 and H3N2 viruses to avoid humoral immunity is well characterized, relatively little is known about the evolution of innate immune antagonism phenotypes in these viruses. Numerous studies have established that only a small subset of infected cells is responsible for initiating the type I and type III interferon (IFN) response during IAV infection, emphasizing the importance of single cell studies to accurately characterize the IFN response during infection. We developed a flow cytometry-based method to examine transcriptional changes in IFN and interferon stimulated gene (ISG) expression at the single cell level. We observed that NS segments derived from seasonal H3N2 viruses are more efficient at antagonizing IFN signaling but less effective at suppressing IFN induction, compared to the pdm2009 H1N1 lineage. We compared a collection of NS segments spanning the natural history of the current seasonal IAV lineages and demonstrate long periods of stability in IFN antagonism potential, punctuated by occasional phenotypic shifts. Altogether, our data reveal significant differences in how seasonal and pandemic H1N1 and H3N2 viruses antagonize the human IFN response at the single cell level.

## Author summary

Successful replication and transmission of influenza A viruses (IAVs) requires effective subversion of the innate immune response. We developed a new single cell method to examine the interferon (IFN) response to IAV infection at the single cell level. We found that the seasonal H3N2 and H1N1 lineages differ significantly in their ability to antagonize IFN signaling and suppress IFN induction, revealing unexpected nuances in how

this study can be found in https://github.com/
BROOKELAB/Influenza-virus-scRNA-seq.

**Funding:** This study was generously supported by
the National Institute of Allergy and Infectious
Diseases of the National Institutes of Health under
awards 1R01AI179910 and 1R01AI139246 to C.B.
B. and 1R01AI165475 to N.C.W. The funders had
no role in study design, data collection and
analysis, decision to publish, or preparation of the
manuscript.

**Competing interests:** The authors have declared
that no competing interests exist.

these lineages interact with the innate immune system. We also directly examined how
the IFN antagonism potential of the viral NS segment has evolved over compared decades
of circulation in humans. Altogether, our data reveal significant differences in the capacity
of the seasonal H1N1 and H3N2 lineages to antagonize the IFN response at the single cell
level over decades of circulation and evolution.

## Introduction

Numerous RNA viruses, including influenza A virus (IAV), are endemic in the human popula-
tion and impose ongoing global public health burdens [1]. The successful emergence, estab-
lishment, and maintenance of a virus in a new host species typically requires effectively
bypassing or antagonizing the activation of innate antiviral defense pathways [2–4]. How
viruses evolve to subvert the innate immune response in humans during and after zoonotic
emergence is a critical aspect of cross-species transmission and adaptation that remains poorly
characterized.

The type I and type III interferon (IFN) signaling pathways play critical roles in restricting
viral replication and spread during the initial stages of infection, and in helping orchestrate
and regulate the adaptive immune response [5–11]. IAV infection is primarily sensed through
recognition of viral RNAs by RIG-I, resulting in the activation of the innate immune response,
including the expression of type I and type III IFN [12]. Secreted IFNs are recognized by type-
specific receptors on both infected and uninfected bystander cells, leading to JAK/STAT-
dependent expression of hundreds of interferon stimulated genes (ISGs) that can collectively
block the replication of diverse viruses [13–15].

Like other viruses, IAV has evolved multiple mechanisms to antagonize and evade the IFN
response [4,16]. The primary innate immune antagonist encoded by IAV is the nonstructural
protein 1 (NS1), expressed by the NS segment [17,18]. NS1 interacts with both RNA and mul-
tiple host proteins to subvert innate immune function via several mechanisms [17,19]. For
example, NS1 can block ubiquitination of RIG-I by TRIM25, preventing RIG-I signaling and
downstream IFN induction [20]. The RNA-binding activity of NS1 can inhibit the antiviral
activity of PKR and OAS [21–23]. NS1 can also more broadly suppress the transcription of
antiviral factors by preventing cellular pre-mRNA processing and nuclear export [24,25].
Beyond NS1, the viral accessory protein PA-X can modulate the innate immune response by
specifically degrading a subset of host mRNAs [26–30].

While the general mechanisms of IAV-mediated immune antagonism are well described,
the extent to which these functions have evolved within the H1N1 and H3N2 lineages over
decades of circulation in humans is not understood. We previously observed that representa-
tives of the seasonal H3N2 and 2009 pandemic H1N1 lineages differ both in IFN induction
potential and the relationship between NS segment expression and IFN/ISG induction at the
single cell level [31]. Other studies have demonstrated significant phenotypic differences
between NS1 genes from human H1N1 and H3N2 isolates [32] and between the pdm2009
H1N1 lineage NS1 and other human NS1 genes [33]. These data strongly suggest that IAV
strains from different years and lineages vary in how they interact with the human innate
immune system.

To better understand how IFN antagonism phenotypes have evolved within the H1N1 and
H3N2 lineages over decades of circulation in humans, we developed a novel flow cytometry-
based fluorescence *in situ* hybridization method to precisely quantify innate immune induc-
tion at the single cell level following infection. We used this approach to reveal that H1N1 and

H3N2 viruses exhibit distinct abilities to antagonize IFN induction versus IFN signaling, and that these differences primarily segregate with the NS segment. Finally, we show how the ability of the current H1N1 lineage NS segment to antagonize IFN induction and signaling has remained largely stable since the 2009 pandemic, while the H3N2 NS segment has undergone significant phenotypic shifts since emerging in 1968. Altogether, our data reveal new dimensions in how IAV antagonizes the IFN response at the single cell level and demonstrate how this capacity evolves over decades of circulation in humans.

## Results

### Quantification of IFN/ISG expression at the single cell level by combining hybridization chain reaction staining with flow cytometry

We and others have previously demonstrated that only a small fraction of IAV-infected cells express type I or type III IFN [31,34–36]. We further showed that different IAV strains can differ in the percentage of infected cells that eventually express IFN [31]. Similar cellular heterogeneity in IFN induction has been observed for other viruses as well [37–41]. Given the rarity of cells driving the IFN response, bulk methods of analyzing the host response may fail to effectively capture important dynamics within these cells. Further, due to the highly non-linear nature of cytokine responses [42,43], even small differences in the percentage of infected cells that produce IFN could have significant effects on downstream infection dynamics. Thus, we aimed to more carefully characterize patterns of single cell innate immune activity during IAV infection.

To examine the effects of infection on host gene expression across millions of individual cells while avoiding the high costs associated with single-cell RNAseq (scRNAseq), low sensitivity and noise associated with this approach [44,45], we adapted hybridization chain reaction-based detection of nucleic acid targets [46] for use with flow cytometry (HCR-flow). To evaluate the sensitivity and specificity of HCR-flow, we measured the expression of the human housekeeping genes *GAPDH* and *ACTB* (as positive controls) and murine *Cd45*(as a negative control) in the human lung epithelial cell line A549. We detected both *GAPDH*, *ACTB* in nearly 100% of cells but not murine *Cd45*, indicating the suitability of HCR-flow specific detection of transcripts within single cells (**S1 Fig**).

We next evaluated the ability of HCR-flow to measure the induction of innate immune-associated transcripts upregulated during viral infection [7,31]. We transfected A549 cells with the RIG-I ligand polyinosinic-polycytidylic acid (pIC) and measured expression of *IFNL1* and the ISG *IFIT3* at different times following transfection using HCR-flow. We observed a rapid increase in *IFIT3* expression in A549s resulting in ~80–90% of cells expressing ISGs by 16–20 hrs. In parallel, we observed a slower increase in *IFNL1* expression that plateaued with only ~20% of the cells expressing detectable levels of *IFNL1* by 20 hrs (**Fig 1A and 1B**). We further observed that *IFNL1* expression frequencies measured by HCR-flow agreed with both IFN transcript levels quantified by bulk qPCR and detection of secreted IFNL in supernatant of pIC treated A549s (**S2 Fig**). These data are consistent with previous reports that IFN induction typically only occurs within a subset of stimulated cells [31,35–37,47,48].

Finally, we assessed the ability of HCR-flow to detect viral infection compared with traditional antibody staining. We infected A549 cells with A/California/07/2009 (Cal07; 2009 pandemic H1N1 strain) and simultaneously stained for viral NP protein with antibody and the viral mRNA that encodes NP with HCR-flow. We observed clear positive signals for both NP protein and NP mRNA that were highly concordant with each other, indicating that HCR-flow performs comparably to antibody staining for detection of IAV infection (**Fig 1C**). Altogether, these data indicated that HCR-flow can be used to quantify the expression of both host and viral RNAs at the single cell level.

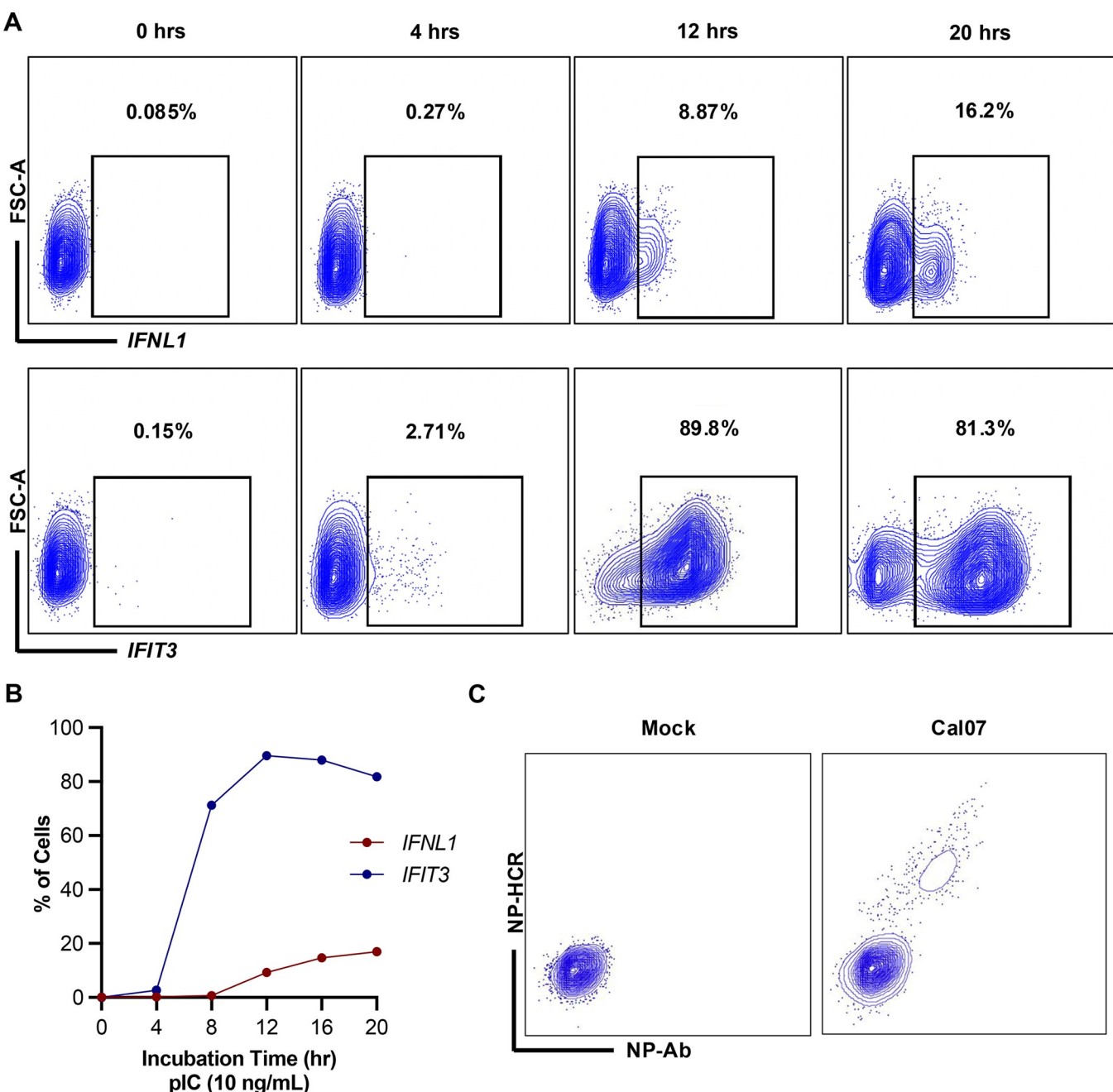

**Fig 1. Adapting hybridization chain reaction (HCR) for quantifying gene expression by flow cytometry (HCR-flow).** (A) Expression of IFNL1 and IFIT3 in A549 cells transfected with pIC 10 ng/mL at different timepoints measured by HCR-flow. (B) Quantification of IFNL1 and IFIT3 expression in A549 treated with pIC (10 ng/mL) and collected at different timepoints. (C) Comparison of infectivity in mock or Cal07 infected (MOI 0.1 NPEU/mL) A549 measured by HCR-flow detecting NP mRNA or traditional antibody staining using HB65 (anti-NP). Data are shown as mean with SD; N = 3 cell culture wells.

## Different IAV subtypes exhibit divergent single cell patterns of IFN and ISG induction

To assess IFN and ISG induction in infected and bystander cells, we infected A549 cells at a multiplicity of infection (MOI) of 0.1 based on NP-expressing units (NPEU) [49] with recombinant Cal07 or A/Perth/16/2009 (Perth09; human seasonal H3N2 isolate) under single cycle

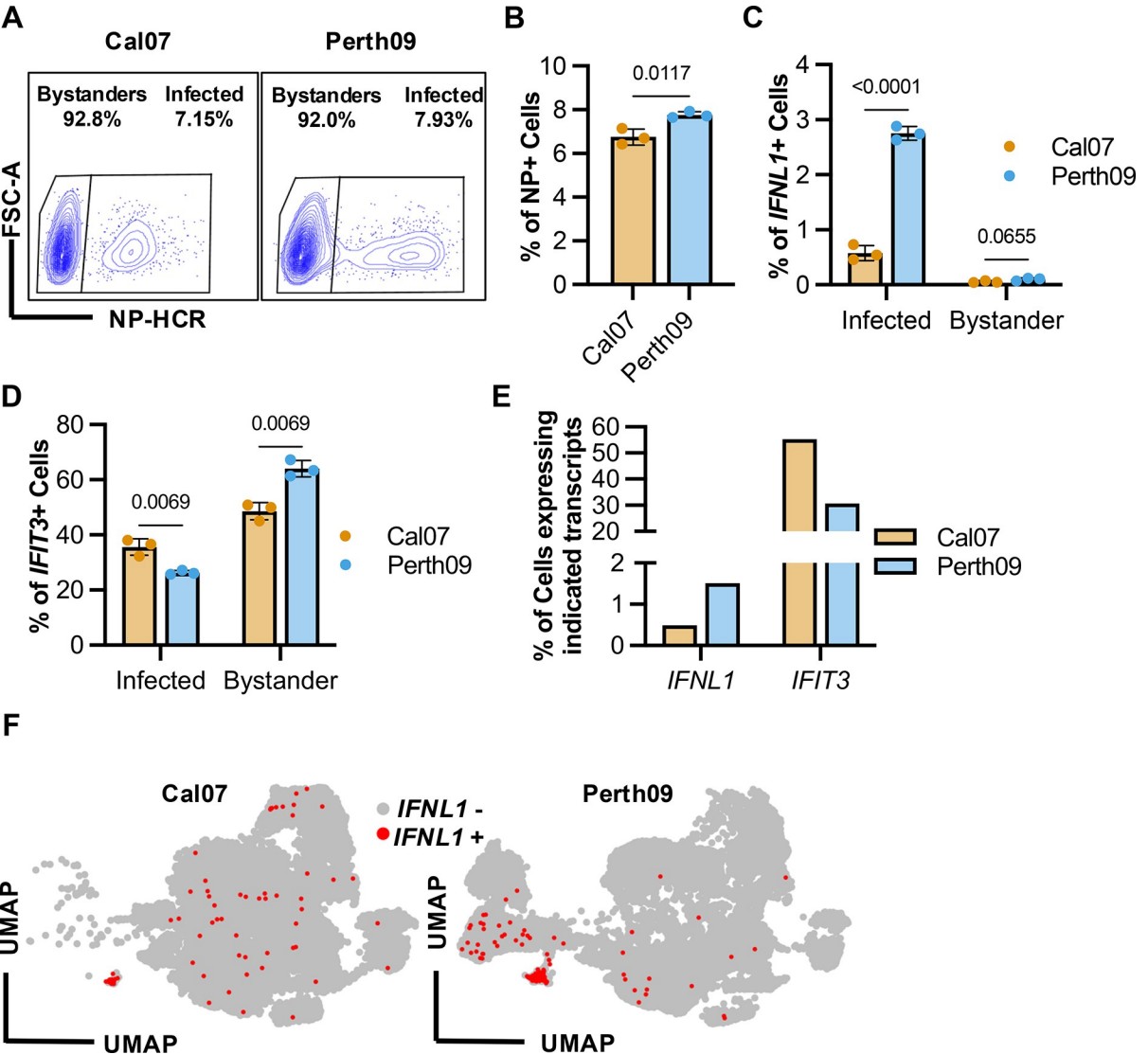

**Fig 2. Differential IFN and ISG antagonism during infection with H1N1 or H3N2 IAV.** (A) Infected A549 cells were separated into bystander or infected based on expression of NP mRNA for detection of IFNL1 and/or IFIT3. (B) Quantification of infectivity by HCR-flow in A549 infected with Cal07 (H1N1) or Perth09 (H3N2) at 16 hpi. (C) Percentage of IFNL1 or (D) IFIT3 positive cells in Cal07 and Perth09 infected and bystander A549 measured by HCR-flow. (E) Percentage of infected cells at 16 hpi with detectable levels of IFNL1 or IFIT3 from scRNA-seq data. (F) UMAP plot from scRNA-seq of A549 infected with either Cal07 or Perth09 highlighting cells with detectable IFNL1 transcript in red. Data are shown as mean with SD; N = 3 cell culture wells with exception of panel (E) which represents a single scRNA-seq library per virus. Multiple unpaired t test (Holm-Šídák method for multiple comparisons) was used for statistical analysis.

conditions. We used a low MOI to ensure that each infected cell was infected with a single virion, as cellular co-infection can alter IFN induction kinetics [50]. At 16 hpi, we collected the cells and examined viral and host gene expression by HCR-flow (**S3 Fig**). Based on detection of viral NP mRNA, we could clearly resolve infected and bystander cells (**Fig 2A and 2B**). We compared expression of *IFNL1* transcript between infected (NP+) and bystander (NP-) cells (**Fig 2C**). We observed that the frequency of *IFNL1* induction in Perth09-infected cells was consistently ~4-5-fold higher than that observed in Cal07-infected cells, consistent with previously published results comparing these strains using scRNAseq [31].

We next assessed induction of the ISG *IFIT3* within infected and bystander cells. In contrast with the *IFNL1* results, a significantly lower percentage of Perth09-infected cells expressed *IFIT3* compared with Cal07-infected cells (**Fig 2D**). At least 50% of all bystander cells expressed *IFIT3*, consistent with widespread paracrine activation of ISG expression and suggesting that the lower levels of *IFIT3* expression in infected cells are due to active suppression by the virus. While the pattern of expression differences between Cal07- and Perth09-infected cells was consistent across three independent experiments, the specific percentages of *IFNL1* + and IFIT3+ cells varied between experiments as a function of cell passage number used. As a result, in this study we are showing a single representative of three independent experiments.

We next compared our HCR-flow results with scRNA-seq data obtained from A549 cells infected with either Cal07 or Perth09. We infected A549 with either Cal07 or Perth09 at MOI of 0.1 under single cycle conditions using ammonium chloride ($NH_4Cl$). We sorted cells at 8 and 16 hpi into bystander or infected populations based on the expression of viral HA protein (**S4 Fig**). We calculated percentages of infected cells with any detectable level of *IFNL1* or *IFIT3* in our scRNA-seq dataset and observed similar patterns of IFN/ISG expression using both single cell methods (**Fig 2E and 2F**). Furthermore, we observed similar differences between the strains in the induction of type I IFNs and larger group of diverse ISGs (**S5 Fig**). Additionally, we compared *IFNL1* counts with either viral NP or NS counts for Cal07 and Perth09 libraries (**S6 Fig**). Perth09-infected cells with high *IFNL1* tended to have lower NS counts while the opposite was observed for Cal07-infected cells. The relationships between *IFNL1* and NP counts were similar for both viruses.

Finally, we compared innate immune activation kinetics between Cal07 or Perth09 by infecting cells and harvesting at different times following infection. We measured bulk *IFNL1* expression by qPCR and secretion of functional type III IFN in the supernatant (**Fig 3A and 3B**). Perth09 infection was associated with faster IFN induction compared with Cal07. The frequencies of *IFNL1+* cells as measured by HCR-flow were consistent with bulk measurements (**Fig 3C**). *IFIT3* induction during Cal07 infection was delayed relative to Perth09; however, it accelerated after 12 hpi and surpassed Perth09-infected cells by 16 hpi (**Fig 3D**). This dynamic could reflect more potent inhibition of IFN signaling by Perth09. NP and NS1 mRNA expression kinetics were largely similar between Cal07 and Perth09, suggesting that their distinct *IFNL1* and *IFIT3* expression phenotypes are not simply the effect of differences in viral gene expression kinetics (**Fig 3E and 3F**). Altogether, these data indicate that Perth09 is more effective at suppressing ISG induction but less effective at avoiding IFN induction, compared with Cal07.

## Strain differences in ISG antagonism are associated with the viral NS segment

The viral NS segment encodes NS1, which plays a well-described role in modulation and antagonism of the innate immune response [51,52]. The NS segments encoded by the 2009pdm H1N1 and H3N2 lineages have distinct natural histories and sequence features and thus potentially differ in their immune antagonism effectiveness. While the Perth09 strain used here is part of the seasonal H3N2 lineage that has circulated in humans since emerging through the 1968 pandemic, the H3N2-associated NS segment has actually circulated in humans continuously since the 1918 pandemic [53]. This is presumably due to reassortment events that allowed the 1918-origin NA segment to persist through the subtype replacements associated with the 1957 H2N2 and 1968 H3N2 pandemics [53–55]. In contrast, Cal07 was isolated during the 2009 pandemic and encodes a classical swine-origin H1N1 NS segment (itself originally derived from the 1918 pandemic virus) that had circulated in swine for decades [55,56]. As a result of these distinct natural histories, Cal07 and Perth09 exhibit limited

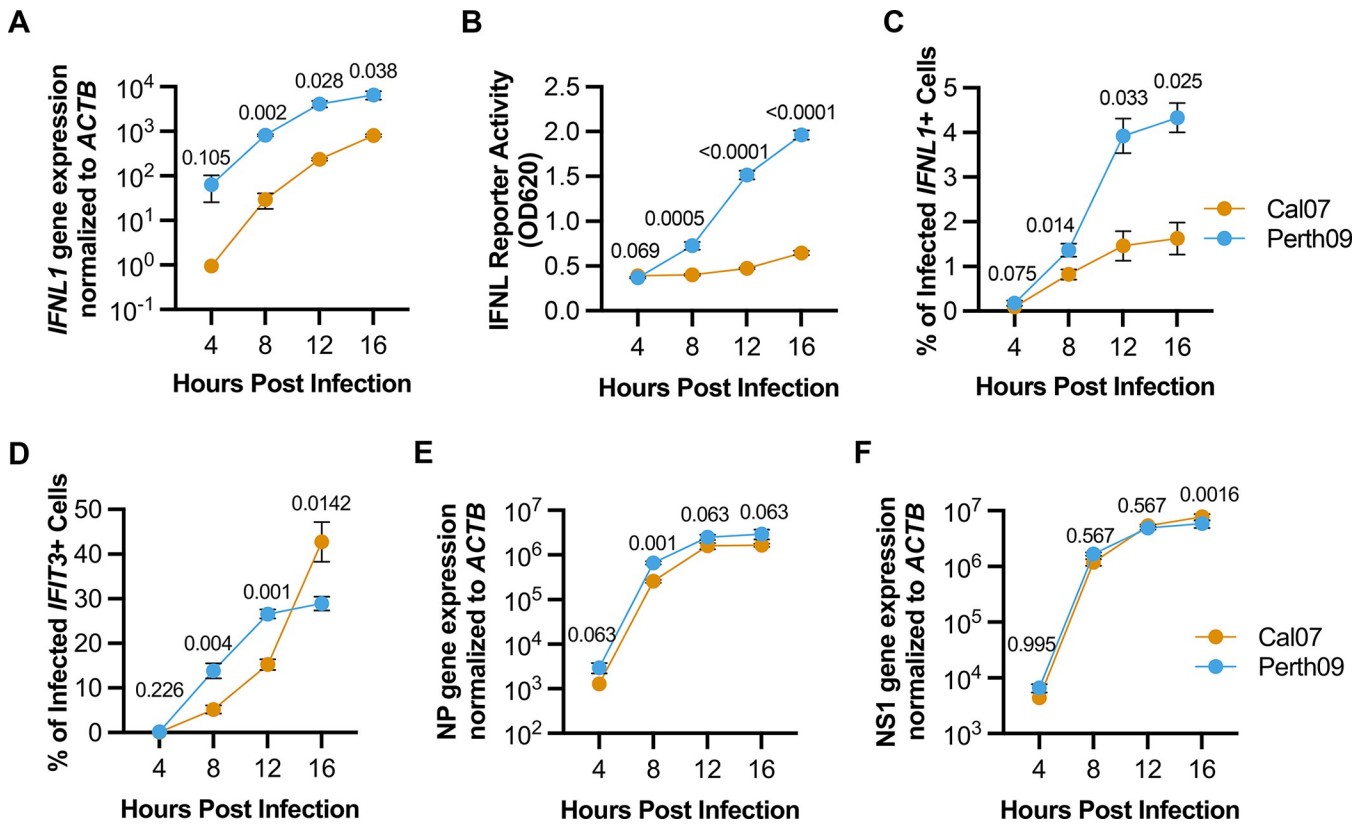

**Fig 3. IFN and ISG induction during infection with H1N1 or H3N2 IAV.** (A) Expression of IFNL1 at different times post infection in A549 infected with Cal07 or Perth09, measured by bulk qPCR. (B) Detection of secreted IFNL in supernatant from A549 infected with Cal07 or Perth09 at different times post infection, measured by an IFNL signaling reporter cell line. (C) Percentage of IFNL1 or (D) IFIT3 positive cells in Cal07 and Perth09 infected A549 at different times post infection, measured by HCR-flow. Expression of NP (E) or NS1 (F) in Cal07 or Perth09 infected cells measured by qPCR. Data are shown as mean with SD with p values indicated on top of each comparison; N = 3 cell culture wells. Multiple unpaired t test (Holm-Šídák method for multiple comparisons) were used for statistical analysis.

nucleotide (81.73%) and amino acid (NS1: 78.08%; NEP: 90.08%) sequence similarity. Based on these differences, we examined the specific contributions of the NS segment to the observed strain differences in IFN/ISG induction.

We generated reassortant viruses in which we swapped the NS segments between Cal07 and Perth09 using reverse genetics (Cal07:NS-Perth09 and Perth09:NS-Cal07). We infected A549 cells at an MOI of 0.1 NPEU/mL under single cycle conditions and collected cells at 16 hpi for HCR-flow analysis. We gated infected based on the presence of detectable NP mRNA signal (**Fig 4A and 4B**). Unlike our results with the parental viruses, we did not observe a significant difference in *IFNL1* expression frequency between the reassortants (**Fig 4C**), suggesting that the different *IFNL1* induction frequency phenotypes observed for Cal07 and Perth09 (**Fig 2C**) are only partially due to the NS segment. In contrast, the *IFIT3* expression phenotype segregated cleanly with the NS segment, as Cal07:NS-Perth09 was associated with lower *IFIT3* expression compared with Perth09:NS-Cal07 (**Fig 4C**). These data demonstrate that the strain differences in ISG expression frequency map to the NS segment.

## Perth09 is more effective than Cal07 at blocking ISG induction

The differences in *IFNL1* and *IFIT3* expression frequencies during Cal07 and Perth09 infection could be explained by differences in either (a) ability to actively suppress IFN/ISG induction

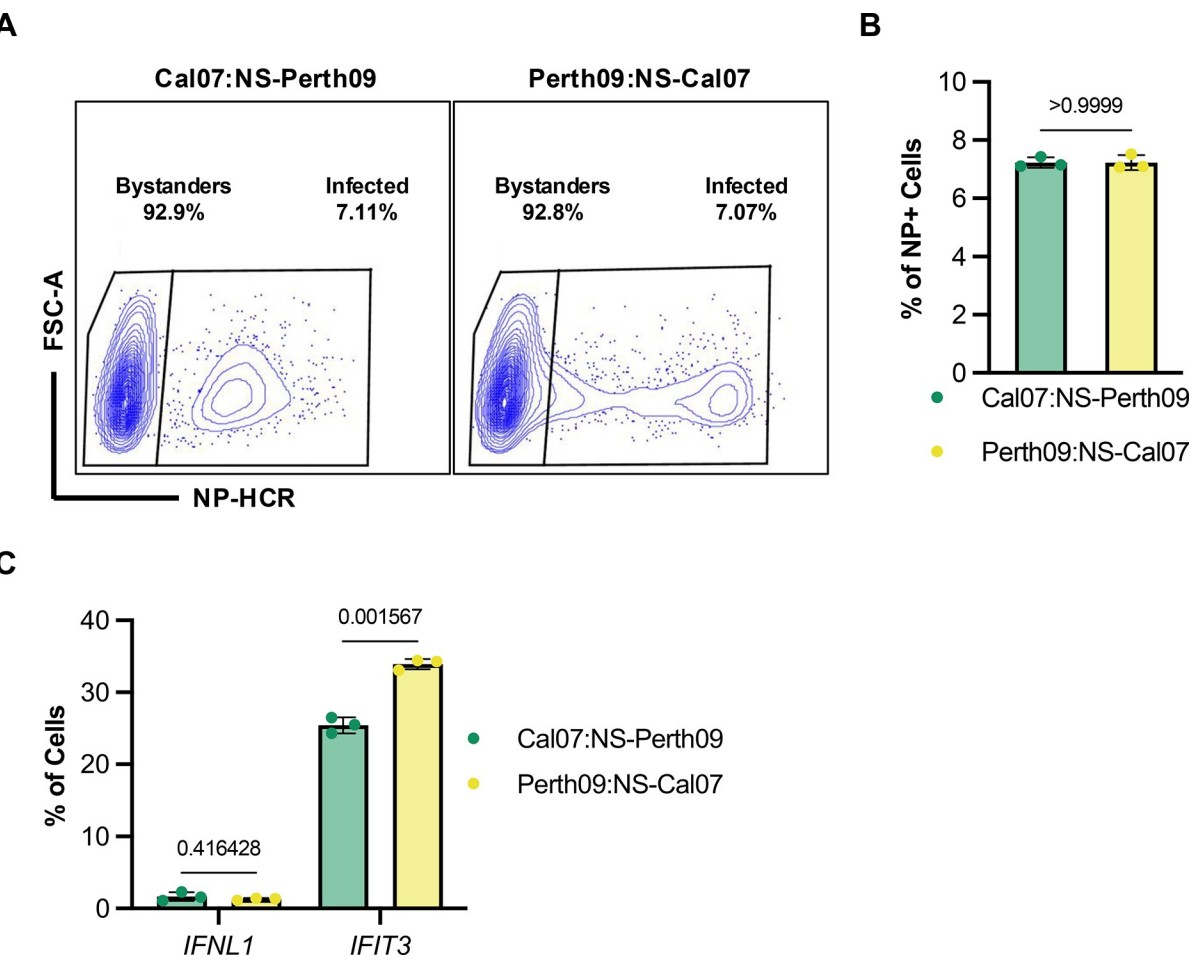

**Fig 4. Differential ISG antagonism observed between H1N1 and H3N2 is dependent on the NS segment.** (A) Infected A549 cells were separated based on the presence or absence of the NP mRNA using HCR-flow. (B) Percentage of infectivity in cells collected at 16 hpi and measured from three replicates for each virus. (C) Quantification of IFNL1 or IFIT3 expression in A549 infected with NS reassortments. Data are shown as mean with SD; N = 3 cell culture wells. Unpaired t test and multiple unpaired t test (Holm-Šídák method for multiple comparisons) were used for statistical analysis.

within infected cells, or (b) ability to evade sensing by the cell, as NS1 has been shown to suppress dsRNA sensing [21,57,58]. To differentiate between these possibilities, we infected A549 cells with either Cal07 and Perth09, treated cells 4 hrs later with 10 ng/mL of the RIG-I agonist pIC, and then assessed *IFNL1* and *IFIT3* expression in mock and infected cells by HCR-flow at 16 hpi.

Uninfected A549 cells transfected with pIC (10 ng/mL) for 16 hrs exhibited relatively high expression frequencies of both *IFNL1* (~20%) and *IFIT3* (~85%), indicating the maximal expression frequencies of these transcripts in the absence of viral interference (**Figs 5A, 5B and S7**). In Cal07 or Perth09 infected cells treated with pIC, *IFNL1* expression frequency was much lower than in mock controls shown in **Fig 5A**, indicating that both strains actively inhibit IFN induction in response to RNA sensing within the cell. Perth09 infected cells consistently exhibited modestly higher rates of *IFNL1* induction compared with Cal07 infected cells, suggesting that Perth09 is slightly less effective at suppressing IFN induction (**Fig 5C**).

*IFIT3* expression frequencies were much lower in Perth09 infected cells versus Cal07 infected cells following pIC treatment (**Fig 5C**). This suggests that Perth09 is more effective

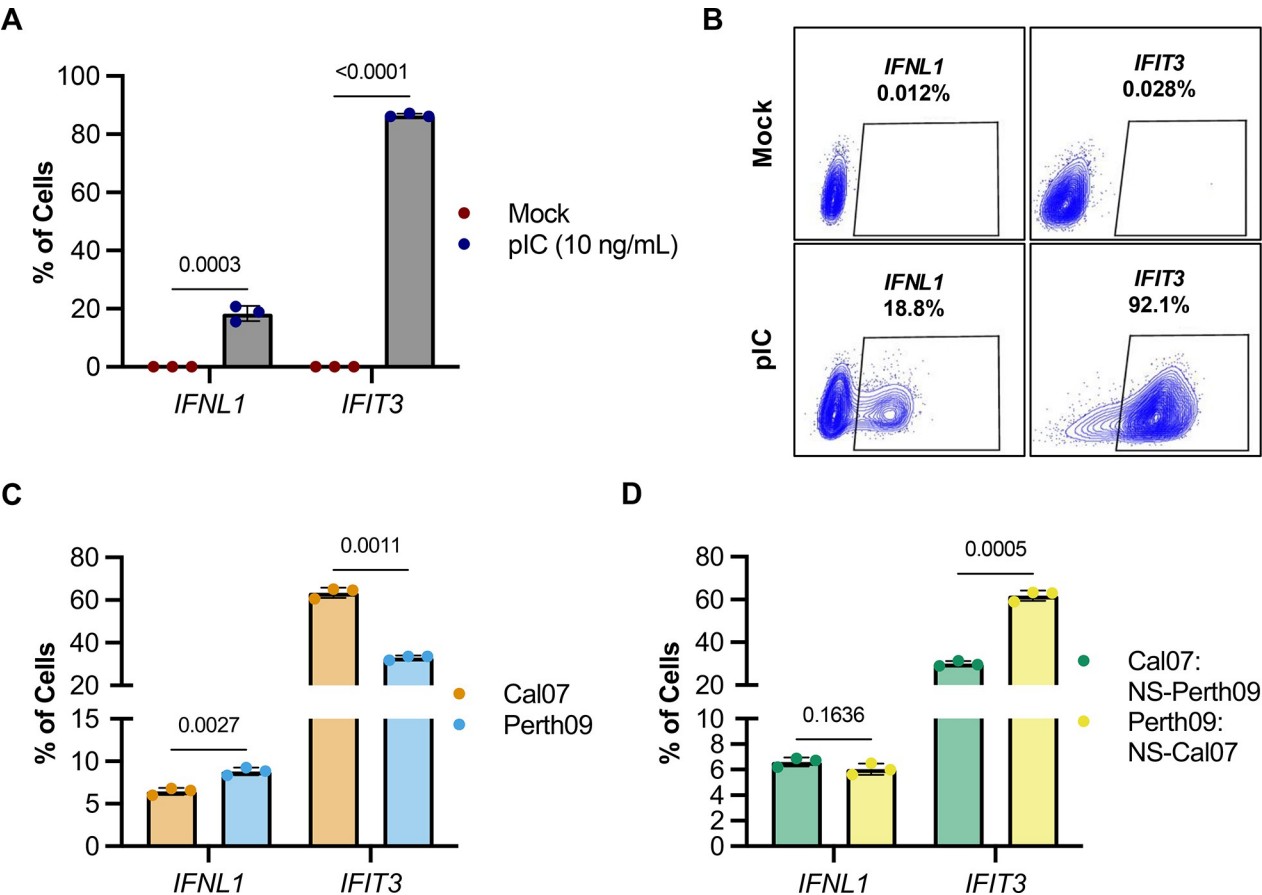

**Fig 5. H3N2 is better at inhibiting ISGs after pIC stimulus compared to H1N1.** (A) Quantification of A549 cells expressing IFNL1 or IFIT3 at 16 hrs post treatment with pIC (10 ng/mL). (B) Percentage of IFNL1 and IFIT3 in mock or pIC (10 ng/mL) treated A549 after 16 hrs. (C) Percentage of IFNL1 and IFIT3 expression in cells infected with Cal07 or Perth09 and subsequently treated with pIC at 4 hpi. (D) Expression of IFNL1 and IFIT3 in A549 cells infected with NS reassortments and treated with pIC measured by HCR-flow. Data are shown as mean with SD; N = 3 cell culture wells. Multiple unpaired t test (Holm-Šídák method for multiple comparisons) was used for statistical analysis.

than Cal07 at suppressing ISG induction rather than simply avoiding ISG induction. Similar to our results with virus-mediated induction of ISG expression, the difference between Perth09 and Cal07 in suppression of pIC-induced ISG expression segregated entirely with the NS segment (**Fig 5D**).

## H3N2 is more effective at suppressing JAK/STAT dependent IFN signaling

Canonical ISG induction occurs downstream of signaling through the type I or type III IFN receptors [59]; however, ISGs can also be directly induced through IRF3 activation, independent of IFN signaling [60]. To determine whether the effects of Perth09 NS on ISG induction resulted from viral antagonism of IFN signaling, we infected A549 cells with either Cal07 or Perth09 and treated the cells with recombinant human IFNB1 or IFNL1 at 4 hpi. We first confirmed that neither IFNB nor IFNL treatment induced expression of *IFNL1* but that both did induce *IFIT3* expression (**Figs 6A and S8**). IFNL1 and IFNB1-driven *IFIT3* expression frequencies were significantly lower in Perth09-infected cells compared to Cal07 (**Fig 6B and 6C**). These data suggest that Perth09 is better at inhibiting type I and type III IFN signaling compared to Cal07.

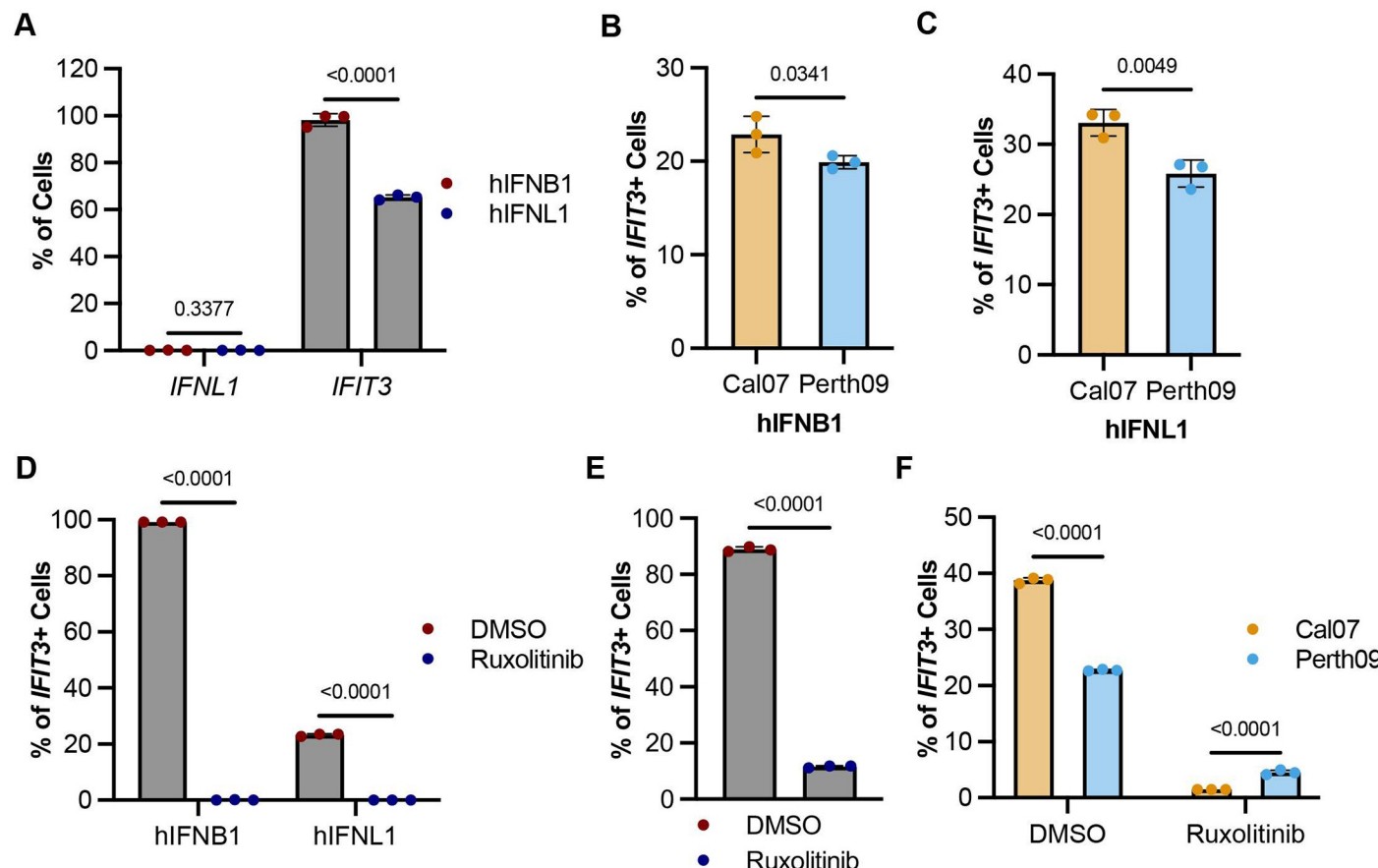

**Fig 6. Perth09 is more effective that Cal07 at antagonizing JAK/STAT signaling.** (A) Percentage of A549 cells expressing IFNL1 or IFIT3 at 16 hrs post treatment with recombinant hIFNL1 or hIFNB1 (100 ng/mL). (B) Comparison of IFIT3 expression in cells infected with Cal07 or Perth09 and subsequently treated with hIFNB1 (100 ng/mL) or (C) hIFNL1 (100 ng/mL). (D) Quantification of IFIT3 frequencies in hIFNB1 or hIFNL1 treated A549 cells with ruxolitinib (10 μM). (E) Percentage of ruxolitinib treated A549 cells expressing IFIT3 at 16 hrs post transfection with pIC (10 ng/mL). (F) Quantification of IFIT3 positive cells in A549s treated with ruxolitinib and subsequently infected with Cal07 or Perth09 for 16 hrs. Data are shown as mean with SD; N = 3 cell culture wells. Unpaired t test and multiple unpaired t test (Holm-Šídák method for multiple comparisons) were used for statistical analysis.

Cells expressing IFNs can upregulate ISGs through autocrine and paracrine IFN signaling, in which type I or type III IFN receptor engagement triggers the canonical JAK/STAT signaling pathway [61]. To investigate if JAK mediated IFN signaling is essential for the differential ISG antagonism between Perth09 and Cal07, we pre-treated cells with the JAK1/2 inhibitor ruxolitinib [62]. In the presence of ruxolitinib, we expect to lose all ISG induction triggered by IFN signaling. To test this, we treated cells with 10 μM ruxolitinib and then exposed them to recombinant IFNB1 or IFNL1 (**Fig 6D**). Under these conditions, ruxolitinib pretreatment completely prevented *IFIT3* induction by recombinant IFNB1 (**S9 Fig**). To quantify IFN signaling-independent induction of *IFIT3* by pIC, we pre-treated cells with 10 μM ruxolitinib, transfected with 10 ng/mL pIC, and then measured the percentages of cells that turned on *IFIT3* expression (**Fig 6E**).

Next, we asked how inhibition of JAK/STAT signaling affected ISG induction during viral infection. As expected, controls pretreated with DMSO prior to Cal07 or Perth09 infection induced *IFIT3* at frequencies similar to what we observed previously (**Fig 6F**). Pretreatment with ruxolitinib flipped the pattern of *IFIT3* expression between Cal07 and Perth09, however,

with a significantly higher percentage of Perth09-infected cells expressing *IFIT3* compared with Cal07. Thus, in the presence of ruxolitinib, the *IFIT3* induction phenotypes of Perth09 and Cal07 closely mirror their *IFNL1* induction phenotypes in the absence of ruxolitinib. These data suggest that Perth09 is less effective than Cal07 at suppressing initial IRF3-mediated induction of IFN and ISGs, but more effective at antagonizing JAK/STAT-dependent ISG induction downstream of IFN signaling.

## Evolution of innate immune antagonism capacity during circulation in humans

Overcoming innate immune defenses is thought to be one of the major factors limiting cross-species transmission [3,63]. Recent evidence from the SARS-CoV-2 pandemic has suggested that viruses may undergo selection for enhanced innate immune antagonism in humans following zoonotic emergence [64]. It is not clear whether seasonal IAVs have undergone similar selection to optimize innate immune antagonism phenotypes as they have adapted to humans over the decades.

We asked whether the single cell IFN/ISG induction phenotypes associated with the seasonal H3N2 and H1N1 lineage NS segments have evolved from their original pandemic emergence and subsequent years of human circulation. We used reverse genetics to generate recombinant viruses with NS segments from representative H3N2 isolates from 1968 to 2020 in the Perth09 background, and H1N1 isolates from 2009 to 2020 in the Cal07 background (Table 1). Representative H3N2 strains from 1968–1999 were chosen from vaccine strains. Representative H1N1 and H3N2 from 2012–2020 were selected from years where significant numbers of new coding substitutions appeared in the NS1 consensus sequence, specifically looking at years with more than five coding changes compared to Perth09 and accumulation of at least one substitution between them. For each year chosen, we selected a representative strain that exhibited high NS1 sequence similarity to the consensus sequence which were generated using available NS1 sequences from each year in the bacterial and viral bioinformatics resource center (BV-BRC) database.

We infected A549 cells at an MOI of 0.1 NPEU/mL with each recombinant virus under single cycle conditions and measured the percentages of infected cells expressing *IFNL1* and/or *IFIT3* using HCR-flow. To ensure fair comparison between the different viral genotypes, we confirmed that infected cell percentages were comparable for all viruses tested (Fig 7A and 7B). We first examined the effects of different historical H3N2-origin NS segments (1968, 1979, 1999, 2009, 2012, 2015, 2018, and 2020) on single cell *IFNL1* and *IFIT3* induction frequencies (Fig 7C and 7E). To account for technical variation between experiments, we normalized everything to the mean values measured for Perth09. We consistently observed that *IFNL1* and *IFIT3* induction frequencies were largely constant from 1968 through 1999,

**Table 1. Virus strains selected for this study with their corresponding NCBI accession number.**

| H1N1 | NCBI Accession Number | H3N2 | NCBI Accession Number |
|---|---|---|---|
| A/California/7/2009 | CY121684.1 | A/HongKong/1/68 | AF348201.1 |
| A/Texas/22/2012 | KC891536.1 | A/Bangkok/1/1979 | CY121004.1 |
| A/California/35/2015 | KT836439.1 | A/Moscow/10/1999 | CY121377.1 |
| A/Brisbane/2/2018 | OQ718997.1 | A/Perth/16/2009 | KJ609210.1 |
| A/Arizona/16/2020 | MT499610.1 | A/Maryland/32_11/2012 | MW790252.1 |
| | | A/California/101/2015 | KX413358.1 |
| | | A/Arkansas/01/2018 | MH126445.1 |
| | | A/Colorado/01/2020 | MT245390.1 |

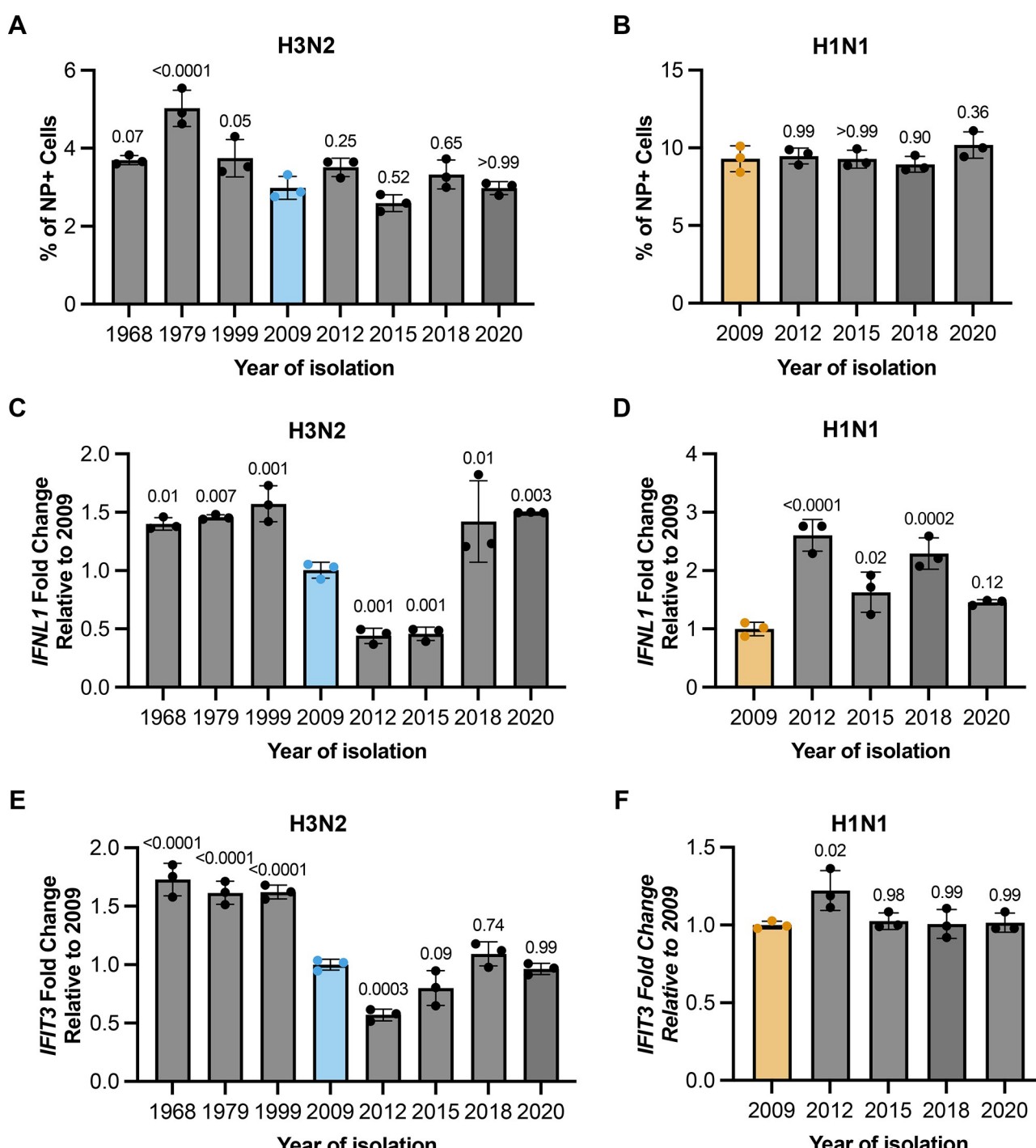

**Fig 7. H3N2 and H1N1 immune antagonism is highly variable during circulation.** Percentage of NP positive cells for (A) H3N2 with NS from 1968–2020 or (B) Cal07 expressing NS from 2009–2020. (C) Quantification of IFNL1 expression in A549s infected with H3N2 encoding NS from 1968–2020 at 16 hpi normalized to 2009. (D) Percentages of IFNL1 positive populations in cells infected with H1N1 expressing NS from 2009–2020 at 16 hpi normalized to 2009 (E) Quantification of IFIT3 in A549s cells infected with H3N2 encoding NS from 1968–2020 at 16 hpi normalized to 2009. (F) IFIT3 expression in A549s infected with H1N1 NS from 2009 to 2020 at 16 hpi normalized to 2009. Data are shown as mean with SD with p values indicated on top for comparison to 2009; N = 3 cell culture wells. One-way ANOVA (Dunnett's Multiple Comparisons test) was used for statistical analysis to compare conditions to 2009.

suggesting that the effect of the NS segment on IFN/ISG induction phenotypes was largely unchanged over the first 30+ years of H3N2 circulation.

Starting with Perth09, *IFNL1* and *IFIT3* induction frequencies decreased through 2015 before increasing again significantly by 2020. These data suggest that the ability of the H3N2-origin NS segment to antagonize IFN and ISG induction has fluctuated significantly at times since the 1968 pandemic; however, there has not been a constant trend towards increased antagonism potency over 50+ years of human circulation.

For the pdm2009 H1N1 lineage NS segment, all post-2009 representatives tested except for A/Arizona/16/2020 exhibited significant increases in *IFNL1* induction frequencies, compared with Cal07 (**Fig 7D**). In contrast, *IFIT3* induction frequencies were largely stable from 2009 to 2020 (**Fig 7F**). These data suggest that, if anything, the pdm2009 H1N1 lineage NS segment has evolved to become less capable of inhibiting IFN induction since its original emergence.

Finally, we examined whether the differences in *IFNL1* and *IFIT3* induction across the different H1N1 and H3N2 isolates tested could be simply explained by changes in relative levels of NS1 expression. We measured NS1 transcript levels in A549 cells infected with either H1N1 or H3N2 at MOI 0.1 by qPCR and normalized to NP mRNA levels to account for any differences in infection kinetics (**S10A and S10B Fig**). We observed a steady increase in relative NS1 expression levels as a function of year of isolation for the H3N2 lineage but not for the pdm2009 H1N1 lineage. When we directly compared relative NS1 expression levels and IFN/ISG expression phenotypes across the different viruses in our panel, we observed a significant negative correlation ($p = 0.0034$) between NS1 expression levels and *IFIT3* induction frequencies for H3N2 viruses, but not for H1N1 (**S10C and S10F Fig**). This suggests that some of the changes in IFN signaling antagonism observed across the H3N2 lineage may be driven by changes in the kinetics and/or magnitude of NS1 expression. No significant correlation was observed between NS1 expression levels and *IFNL1* induction frequencies for either lineage.

Altogether, our analyses suggest that the innate immune antagonism phenotypes of the seasonal H1N1 and H3N2 NS segments have varied significantly over the years; however, neither lineage has exhibited a consistent trend of evolving increasingly potent antagonism of IFN or ISG induction since emerging into humans.

## Discussion

Four distinct lineages of IAV (H1N1:1918–1957 and 1976–2009; H2N2:1957–1968, H3N2:1968-present; and 2009pdm H1N1:2009-present) have circulated in humans over the past century. The survival of these lineages in the human population depends upon the continual emergence of antigenic escape variants that facilitate reinfection of individuals with pre-existing humoral immunity [54,65,66]. While viral antigenic evolution is well studied, the evolution of innate immune suppression within and between the human seasonal IAV lineages is less well understood. To help address this knowledge gap, we developed a novel method for quantifying transcriptional changes within single cells to precisely compare the relative abilities of the current H1N1 and H3N2 lineages to suppress IFN induction and signaling.

We found that the H3N2 lineage is less effective at avoiding IFN induction than H1N1, but more effective at suppressing JAK/STAT-dependent IFN signaling, and that these phenotypes track largely but not entirely with the NS segment. Additionally, we compared a panel of NS segments from across the human circulation histories of the H3N2 and 2009pdm H1N1 lineages and observed substantial variation within the lineages but no clear trend of increasing immune antagonism efficiency in either lineage. These data suggest that while the H3N2 and 2009pdm H1N1 lineages clearly differ in IFN antagonism phenotypes, neither lineage appears to have been under consistent strong selection to increase their IFN antagonism ability,

suggesting that both lineages were already sufficiently capable of antagonizing the human IFN system when they emerged into humans. This is not entirely surprising for the H3N2-origin NS segment, which may have been in human circulation since 1918, but is more surprising for 2009pdmH1N1, which picked up its NS segment from swine origin viruses [53,67]. These data are also consistent with the idea that selection for evasion of adaptive immunity is likely far stronger in humans than selection to maximize antagonism of innate immunity.

While our data suggest that selection for enhanced IFN antagonism is a not a major driver of IAV evolution in humans, this does not mean that it does not impose a significant barrier on spillover and initial spread of zoonotic strains. Despite the absence of a simple trend towards enhanced IFN antagonism, significant evolutionary shifts have occurred in IFN antagonism ability, at least in the H3N2 lineage. More work is needed to determine how the phenotypic shifts we observed (*e.g.* H3N2 viruses between 2009 and 2012) relate to broader evolutionary trends at the host population scale. For instance, it is possible that NS segments with increased or decreased IFN antagonism potential could piggyback along with an antigenically novel HA variant that sweeps during an antigenic cluster transition.

In addition to NS1, the NS segment encodes a second protein, nuclear export protein (NEP) through mRNA splicing. NEP facilitates the nuclear export of viral genomic RNAs and has been implicated in antagonizing the IFN-associated transcription factor IRF7 [68–70]. Comparisons of NEP sequences across the H1N1 and H3N2 viruses tested showed no relationship between amino acid substitutions in NEP and phenotypic changes in IFN/ISG antagonism, suggesting that NEP is unlikely to significantly contribute to the differences we observed.

The IFN response is a highly nonlinear process, meaning that small differences during the initiation phase can have disproportionately large effects on the magnitude of the downstream response [42]. This is because each individual cell that secretes IFN can trigger the defenses and enhance the IFN production of potentially hundreds of additional cells. Thus, relatively modest differences in the numbers of cells capable of inducing IFN (as we observe between H1N1 and H3N2 viruses) may have large effects on subsequent infection dynamics [71]. Based on this, we hypothesize that the percentage of infected cells capable of expressing and secreting IFN is an important parameter of the overall infection response which cannot be measured accurately by bulk methods. Single cell approaches like those we use here are essential both for precisely quantifying critical features of host-virus interactions and for directly examining the rare events at the cellular scale that can have disproportionate effects on infection dynamics at the organismal scale.

Altogether, our results shed new light on how the seasonal H1N1 and H3N2 lineages differ in their interactions with the human IFN system and detail how these phenotypes have evolved over decades of human circulation. These differences in IFN antagonism may contribute to observed differences in infection severity associated with H3N2 versus H1N1 [72], as well as influence other differences in within- and between-host dynamics of the two subtypes [55]. Taken together, our data demonstrate that human-adapted influenza strains differ in their ability to modulate specific features of the IFN response and suggest that more work is needed to account for natural viral genetic variation in understanding IAV interactions with the host.

## Materials and methods

### Plasmids and cell lines

A/California/04/09 plasmids were provided by Dr. Jonathan Yewdell, and substitutions A660G and A335G were introduced into HA and NP, respectively, to convert them into A/California/07/09 (NCBI accession numbers CY121680 and CY121683). Dr. Seema Lakdawala

generously provided the plasmids encoding A/Perth/16/09 viral segments. Madin-Darby canine kidney cells (MDCK) and human lung epithelial cells (A549) were both obtained from Jonathan Yewdell. MDCK and A549 cells were maintained in Gibco's minimal essential medium with GlutaMax (Life Technologies) and Gibco's Dulbecco's Modified Eagle Medium high glucose supplemented with GlutaMax and sodium pyruvate (Life Technologies), respectively. Human kidney embryonic cells (293T) were provided by Dr. Joanna Shisler and maintained in Gibco's minimal essential medium with GlutaMax (Life Technologies). All cell lines were supplemented with 8.3% fetal bovine serum (Avantor) and grown at 37˚C with 5% CO2.

## Viruses

Recombinant A/California/07/09 and A/Perth09/16/09 were rescued using a reverse genetics system by transfecting 60%-80% confluent 293T with 500ng of each plasmid using jetPRIME (Polyplus). MDCK cells were used to amplify the virus rescue and to generate working stock by infecting T-75 flasks at MOI of 0.01 based on TCID50/mL. Sequences for NS segments selected from historic strains (**Table 1**) were synthesized using gBlocks (IDT) and cloned into the pDZ dual-promoter reverse genetics vector using SapI sites. Constructs were used to rescue viruses using the reverse genetic system described previously.

## Viral infections

Recombinant viruses were titered by infecting A549 under single cycle conditions (imposed by adding neutralizing antibodies EM4-C04 [2.75 µg/ml; for Cal07] or CO5 [6.5 µg/ml; for Perth09] to cell supernatant after binding and entry) and quantifying infected cell numbers by flow cytometry using the anti-NP monoclonal antibody HB65. By combining viral dilution factor and the number of infected cells, we were able to calculate NP-expressing units/mL (NPEU/mL) as previously described [73]. A549 were infected at MOI of 0.1 NPEU/mL by diluting virus in PBS + 0.1% BSA and incubating for 1 hr at 37˚C, after incubation the monolayer was rinsed with PBS and growth media supplemented with 8.3% FBS and neutralizing antibodies EM4-C04 [2.75 µg/ml] or CO5 [6.5 µg/ml] to block viral spread and cells were collected at 16–18 hpi. Infectivity was tested by staining for viral protein NP using the anti-NP mouse monoclonal HB65. Cells were incubated with Foxp3/Transcription Factor Staining Buffer Set (eBioscience) for 30 mins at 4˚C and rinse twice with PBS supplemented with 0.1% BSA and 0.1% Saponin before staining. Cells were incubated with HB65 conjugated to Alexa Fluor 647 diluted in PBS supplemented with 0.1% BSA and 0.1% Saponin for 30 mins at 4˚C. After incubation cell were rinsed with PBS twice and analyze using BD FACSymphony A1.

Infections for scRNA-seq experiment were performed in A549 cells incubated with virus diluted in PBS + 0.1% BSA for 1 hr at 37˚C at MOI of 0.1 based on NPEU/mL. After incubation, the cells were rinsed with PBS, and growthy media was added. At 3hpi, growth media was removed, and F-12 was supplemented with 50 mM HEPES and 20 mM $NH_4Cl$ to block secondary spread. After incubation, cells were rinsed with PBS stained with HA stem-specific antibody FI6v3 and washed before being sorted into mock, bystander, and infected populations using a Bigfoot cell sorter.

## Single cell RNA-seq analysis

Sorted A549s were counted, and viability was determined on a BD20 cell counter (BIO-RAD). Cells were diluted to desired concentrations (5000 cells/sample) and individual cDNA libraries were made using the 10x Chromium Single cell 3'(Pleasanton, CA) as per manufacturers protocol. Following cDNA synthesis, libraries were sequenced on an Illumina NovaSeq 6000

using S4 flowcells. In house analysis used for this study can be found in https://github.com/BROOKELAB/Influenza-virus-scRNA-seq.

## Polyinosinic-polycytidylic acid (pIC) treatment

Concentration of polyinosinic-polycytidylic acid (pIC) (Invivogen: tlrl-picw) was determine by transfecting A549 using lipofectamine 3000 transfection reagent (Invitrogen) following manufacturer's instructions. After 16 hrs RNA was isolated using RNeasy Mini kit (Qiagen) and cDNA synthesis was performed using Verso cDNA synthesis kit (Thermo Scientific). Reaction were setup by adding 4 μL of 5x cDNA synthesis buffer, 2 μL of dNTPs, 1 μL Oligo-dT, 1 μL of RT enhancer, 1 μL of Verso enzyme, 5 μL of cDNA and 6 μL of nuclease-free water. Mixture was incubated at 45˚C for 50 min and 95˚C for 2 min and held at 4˚C; cDNA was stored at -20 and subsequently used for quantitative pcr. TaqMan fast advanced master mix (Applied Biosystems) was used to carry out the qPCR in a QuantStudio 3 (Applied Biosystems). Reaction was setup as follows 10 μL of TaqMan fast advanced MM, 1 μL of *IFNL1* probe (Applied Biosystems: Hs00601677_g1), *IFIT3* probe (Applied Biosystems: Hs01922752_s1) or *ACTB* probe (Applied Biosystems: Hs01060665_g1), 1 μL of cDNA and 8 μL of nuclease-free water. Cycle conditions for qPCR were setup as 50˚C for 2 min, 95˚C for 2 mins, and 40 cycles of 95˚C for 1 second followed by 60˚C for 20 seconds.

## Quantification of viral mRNA

RNA from infected A549s was isolated and cDNA synthesis was performed as described above. Expression of NS1 and NP was measured by qPCR using PowerUp SYBR Green Master Mix (Applied Biosystems) Reaction was setup as follows 10 μL of using PowerUp SYBR Green MM, 0.5 μL of 10 μM forward and reverse primers (H1-NS1-forward: AACACCCTTGGCCTCGATAT, H1-NS1-reverse: TGAGCATGAACCAGTCTCGT, H1-NP-forward: CCCAGGAAACGCTGAGATTG, H1-NP-reverse: GACCAGTGAGTACCCTTCCC, H3-NS1-forward: CCATTCCTTGATCGGCTTCG, H3-NS1-reverse: TCCTTCCATTTTCTGCTTGGG, H3-NP-foward: TCGGGACAATGGTGATGGAA, H3-NP-reverse: CCTGGGTTCCGACTTTCTCT), 1 μL of cDNA and 8 μL of nuclease-free water. Cycle conditions for qPCR were setup as 50˚C for 2 min, 95˚C for 10 mins, and 40 cycles of 95˚C for 15 seconds followed by 60˚C for 1 minute.

## Recombinant interferon treatment

Dose of recombinant human IFNB1 and IFNL1 (PeproTech) was determine by diluting IFNs growth media and incubated with A549 cells. After 16 hrs RNA was isolated, cDNA synthesis was performed and IFIT3 expression was measured by qPCR as described previously. Inhibition of JAK 1/2 by Ruxolitinib or Baricitinib (MedChemExpress) was determined by diluting inhibitors in growth media and incubated with A549 for 4 hrs before inducing IFN signaling using recombinant hIFNB1. After 16 hrs post IFN treatment RNA was isolated, cDNA synthesis was performed and *IFIT3* expression was measured by qPCR as described above.

## Detection of secreted IFNL

Supernatant from A549 cells transfected with pIC or infected with IAV was collected and incubated with HEK-Blue IFN-λ reporter cells (Invivogen) for 24 hrs. After incubation, supernatant from HEK-Blue cells was incubated with QUANTI-Blue reagent (Invivogen) as per manufacturers protocol. Presence of IFNL in supernatant was determined by measuring the optical density at 620 nm.

### Hybridization chain reaction combined with flow cytometry (HCR-flow)

Single cell suspension from A549 cells was prepared by incubating cells with 0.05% Trypsin-EDTA (Gibco) at 37˚C for 5 mins. Cell suspension was washed with DPBS (Gibco) and fixed with 4% formaldehyde (Thermo Scientific) for 30 mins at 4˚C. After incubation, cells were washed twice with DPBS and incubated overnight in 70% ethanol. Single cells suspension was washed twice with DPBS + 0.1% Tween 20 (Thermo Scientific Chemicals) and incubated with amplification buffer (Molecular Instruments) for 30 mins at 37˚C. After incubation, 1-3 µM of each probe (Molecular Instruments) was added to cell suspension and incubated overnight at 37˚C. Cells were resuspended in probe wash buffer (Molecular Instruments) and incubated at 37˚C for 10mins and this was repeated three times. Cells were washed once with 5x sodium chloride sodium citrate + 0.1% Tween 20 (SSCT) and incubated in amplification buffer (Molecular Instruments) at RT for 30 mins. After incubation 3-6 µM of each snap-cooled hair-pin (Molecular Instruments) was added to cells suspension and incubated at RT overnight. Cells were washed twice with SSCT and analyzed using BD FACSymphony A1.

## Supporting information

**S1 Fig. Quantification of genes expression frequencies in A549 cells by HCR-flow.** (A) Flow cytometry quantification of A549 cells stained for human GAPDH, ACTB, and mouse CD45 mRNA using HCR-flow. (B) Percentage of cells expressing housekeeping genes and mCD45 measured by flow cytometry in both cells stained using gene specific probes and matching amplifiers or only amplifiers. Data are shown as mean with SD; N = 3 cell culture wells.
(TIF)

**S2 Fig. Expression and detection of IFNL after pIC transfection.** (A) IFNL1 expression from A549 cells transfected with pIC (10 ng/mL) and collected at different timepoints measured by qPCR. (B) Detection of secreted IFNL in supernatant from pIC transfected A549s at different timepoints. Data are shown as mean with SD; N = 3 cell culture wells.
(TIF)

**S3 Fig. Flow cytometry schematic used to determine expression of IFN/ISG in infected and bystander cells.** Expression of IFNL1 and IFIT3 mRNA in A549 cells infected with Cal07 at MOI 0.1 NPEU using HCR-flow.
(TIF)

**S4 Fig. Flow cytometry schematic used to sort cells for scRNA-seq libraries.** Infected A549s were sorted using FI6v3 (anti-HA) to label infected cells (HA+) or bystander (HA-) prior to preparation of scRNA-seq libraries.
(TIF)

**S5 Fig. Expression of immune associated genes in Cal07 or Perth09 scRNA-seq libraries.** Differences in gene expression between Cal07 or Perth09 infected A549 using scRNA-seq at 16 hpi. Comparison of IFNs (A-B) or ISG (C-F) expression in A549 infected with Cal07 or Perth09.
(TIF)

**S6 Fig. Expression of viral genes and IFNL1 in Cal07 or Perth09 scRNA-seq libraries.** Expression of NP (A) or NS (B) counts compared to IFNL counts across Cal07 and Perth09 libraries.
(TIF)

**S7 Fig. Concentrations of polyinosinic-polycytidylic acid (pIC) tested in A549 cells.** Cells were transfected with pIC to induce immune activation which was quantified by measuring IFNL1 expression. Data are shown as mean with SD; N = 3 cell culture wells.
(TIF)

**S8 Fig. Concentrations of recombinant hIFNL1 and hIFNB1 tested in A549s.** Cells were treated with recombinant IFNs for 16 hrs and ISG induction was measured by quantification of IFIT3 expression. Data are shown as mean with SD; N = 3 cell culture wells.
(TIF)

**S9 Fig. Concentration of ruxolitinib tested in A549 cells.** Inhibition was tested by treating A549 for 4 hrs prior immune activation with hIFNB1 (100 ng/ml) and measured by quantifying IFIT3 expression. Data are shown as mean with SD; N = 3 cell culture wells.
(TIF)

**S10 Fig. Quantification of NS1 across H3N2 and H1N1 and correlation with immune activation.** (A) Ratio of NS1 to NP transcripts in A549 cells infected with H3N2 expressing NS from 1968–2020 or (B) H1N1 expressing NS from 2009–2020 at MOI of 0.1 based on NPEU and measured by qPCR. (C) Correlation between IFNL1 fold change and ratio of NS1/NP transcripts in A549 infected with H3N2 expressing NS from 1968–2020 or (D) H1N1 expressing NS from 2009–2020 at MOI 0.1 based on NPEU.(E) Correlation between IFIT3 fold change and ratio of NS1/NP transcripts in A549 cells infected with H3N2 expressing NS from 1968–2020 or (D) H1N1 expressing NS from 2009–2020 at MOI 0.1 based on NPEU. Data are shown as mean with SD with p values indicated on top for comparison to 2009; N = 3 cell culture wells. One-way ANOVA (Dunnett's Multiple Comparisons test) was used for statistical analysis to compare conditions to 2009 and correlation coefficient and p value was determined by calculating Pearson correlation.
(TIF)

## Acknowledgments

We are grateful for Dr. Jenny Drnevich from HPCBio and DNA Services core within the Roy J. Carver Biotechnology Center for the support preparing scRNA-seq libraries and initial data analysis.

## Author Contributions

**Conceptualization:** Joel Rivera-Cardona, Christopher B. Brooke.

**Funding acquisition:** Nicholas C. Wu, Christopher B. Brooke.

**Investigation:** Joel Rivera-Cardona, Neeha Kakuturu, Elizabeth F. Rowland, Qi Wen Teo, Elizabeth A. Thayer, Timothy J. C. Tan, Jiayi Sun.

**Methodology:** Joel Rivera-Cardona, Elizabeth F. Rowland, Collin Kieffer.

**Project administration:** Christopher B. Brooke.

**Supervision:** Collin Kieffer, Nicholas C. Wu, Christopher B. Brooke.

**Visualization:** Joel Rivera-Cardona.

**Writing – original draft:** Joel Rivera-Cardona, Christopher B. Brooke.

**Writing – review & editing:** Joel Rivera-Cardona, Christopher B. Brooke.

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
