## [Decision Letter · Decision Letter 0]

19 Sep 2024

Dear Dr Brooke,

Thank you very much for submitting your manuscript "Seasonal influenza A virus lineages exhibit divergent abilities to antagonize interferon induction and signaling" for consideration at PLOS Pathogens. As with all papers reviewed by the journal, your manuscript was reviewed by members of the editorial board and by several independent reviewers. The reviewers appreciated the attention to an important topic. Based on the reviews, we are likely to accept this manuscript for publication, providing that you modify the manuscript according to the review recommendations.

Sincerely,

Jacob S. Yount

Academic Editor

PLOS Pathogens

Kanta Subbarao

Section Editor

PLOS Pathogens

Michael Malim

Editor-in-Chief

PLOS Pathogens

orcid.org/0000-0002-7699-2064

Reviewer Comments (if any, and for reference):

Reviewer's Responses to Questions

**Part I - Summary**

Reviewer #1: This manuscript by Rivera-Cardona uses a clever new technique and series of recombinant viruses to evaluate the differential interferon induction observed across influenza virus strains. Overall the study is well designed and executed. The new approach described here was rigorously evaluated and will be of great interest to the field. This disparities in IFN antagonism across strains and time is interesting and may shed light on evolutionary trajectories of new flu viruses as they enter the human population. I only have minor concerns.

Reviewer #2: This manuscript by Rivera-Cardona et al investigates the ability of seasonal influenza A viruses to antagonize the induction of interferon (IFN) and IFN-stimulated genes (ISGs). While the laboratory has previously shown relatively few IAV-infected cells express type I or type III IFN, how this response, and IFN antagonism has changed through evolution of H1N1 and H3N2 strains circulating in human populations over time has remained relatively unknown. To address this question, the authors developed a sensitive HCR-flow method to determine relative abundance of ISG (IFIT3) and IFN (IFNL1) transcripts and influenza NP in single cells. As controls for the method, the authors show correlation between this method and NP antibody staining, scRNA-seq, and qPCR. This flow cytometric-based FISH developed by the authors will be useful for many others studying not only influenza virus infection, but innate immune and IFN responses. The authors demonstrate variability in ISG and IFN induction between H1N1 and H3N2 viruses and, within each, changes in the ability to induce/antagonize these responses throughout years of circulation in humans. This is a well written and executed study, but minor changes and perhaps a small amount of additional experimental detail would allow for better interpretation of the results presented.

**Part II – Major Issues: Key Experiments Required for Acceptance**

Reviewer #1: None noted

Reviewer #2: Are there kinetic differences in the induction of ISG and IFN between Cal07 and Perth09? While a timecourse is shown for the induction of responses to polyI:C, this is not investigated for either infection. Based on Figure S4, there is more HA present in Cal07 at 8 and 16hrs, and potentially earlier replication, which could impact induction and antagonism of the ISG responses. In Figure 2, it appears as though there may be slightly more NP-HCR following Perth09 at the timepoint analyzed. Whether differential changes in infection kinetics independent of the NS1 segment could partially contribute to changes in responses is not addressed or discussed.

**Part III – Minor Issues: Editorial and Data Presentation Modifications**

Reviewer #1: For Fig 2 it may be helpful to show the NP and IFNL on the same flow plot to visualize potential correlations between virus levels and IFNL across individual cells. Addition of NS1 would further strengthen this analysis but may be beyond the scope of this manuscript.

Fig. S9 is pretty important to evaluate Fig. 6. Can this be moved into the main figure?

Line 43 - potential typo, assuming it should be NS not NA

Line 103 - Several other groups have also demonstrated heterogeneity of virus-induced IFN induction and should be cited (PMIDs 30626670, 33507952, 33730024).

Line 111 - the authors discuss the high cost of single cell, is it worth also noting the low sensitivity and stochasticity? Which may be overcome by their approach?

Number of samples/replicates is missing for some data

Reviewer #2: How many times each experiment was performed and individual replicates (or error bars) are not included for several figure panels throughout and should be added.

Statistics and data plotted are based on culture wells analyzed. Were experiments repeated multiple times? If so, showing replicates (even in a supplemental figure) would be helpful to assess heterogeneity of the system/assay between infection experiments. At a minimum, potential heterogeneity across infection experiments could be mentioned in the text with data not shown.

It is unclear why statistics are included on some graphs where results are described as different, but not others. Statistical analysis should be updated throughout. Additionally, post-hoc analysis for data in Figure 6 to know which viruses/years have statistically significant changes in IFIT3 relative induction compared to the 2009 isolates.

In Figure 5F, it appears as though Ruxolitinib significantly reduces IFITM3 expression in both viral infections despite more + signal remaining in Perth09-infected wells. This suggests ISG antagonism is at least partially dependent upon JAK/STAT signaling for H3N2. Language should be softened in the manuscript to address this.

PLOS authors have the option to publish the peer review history of their article (what does this mean?). If published, this will include your full peer review and any attached files.

Reviewer #1: No

Reviewer #2: No

Figure Files:

Data Requirements:

Reproducibility:

References:

---

## [Editor Report · Decision Letter 1]

4 Nov 2024

Dear Dr Brooke,

We are pleased to inform you that your manuscript 'Seasonal influenza A virus lineages exhibit divergent abilities to antagonize interferon induction and signaling' has been provisionally accepted for publication in PLOS Pathogens.

Best regards,

Jacob S. Yount

Academic Editor

PLOS Pathogens

Kanta Subbarao

Section Editor

PLOS Pathogens

Michael Malim

Editor-in-Chief

PLOS Pathogens

orcid.org/0000-0002-7699-2064
---

## [Editor Report · Acceptance letter]

14 Nov 2024

Dear Dr Brooke,

We are delighted to inform you that your manuscript, "Seasonal influenza A virus lineages exhibit divergent abilities to antagonize interferon induction and signaling," has been formally accepted for publication in PLOS Pathogens.

Best regards,

Michael Malim

Editor-in-Chief

PLOS Pathogens

orcid.org/0000-0002-7699-2064